# Genetic associations with ratios between protein levels detect new pQTLs and reveal protein-protein interactions

## Graphical abstract

## Authors

Karsten Suhre

## Correspondence

kas2049@qatar-med.cornell.edu

## In brief

Protein quantitative trait loci (pQTLs) are a valuable resource for drug target development. Here, Suhre shows that a systematic analysis of ratios between protein pairs strengthens associations at known pQTL loci by several hundred orders of magnitude (p-gain), suggesting that these ratios reflect biological links between the implicated proteins.

## Highlights

- Use of ratios increased the number of proteogenomic loci by 25%

- Use of ratios reduced the number of Olink targets without a *cis*-pQTL by 13%

- Ratio QTLs are 7.6-fold enriched in established protein-protein interactions

- Theoretical development and generalization of the concept of ratios are warranted

Suhre, 2024, Cell Genomics 4, 100506
March 13, 2024 © 2024 The Author(s).

# Cell Genomics

CellPress

## Article

# Genetic associations with ratios between protein levels detect new pQTLs and reveal protein-protein interactions

Karsten Suhre[1,2,3,*]

[1]Bioinformatics Core, Weill Cornell Medicine-Qatar, Education City, Doha 24144, Qatar
[2]Englander Institute for Precision Medicine, Weill Cornell Medicine, New York, NY 10021, USA
[3]Lead contact
*Correspondence: kas2049@qatar-med.cornell.edu

## SUMMARY

Protein quantitative trait loci (pQTLs) are an invaluable source of information for drug target development because they provide genetic evidence to support protein function, suggest relationships between *cis*- and *trans*-associated proteins, and link proteins to disease endpoints. Using Olink proteomics data for 1,463 proteins measured in over 54,000 samples of the UK Biobank, we identified 4,248 associations with 2,821 ratios between protein levels (rQTLs). rQTLs were 7.6-fold enriched in known protein-protein interactions, suggesting that their ratios reflect biological links between the implicated proteins. Conducting a GWAS on ratios increased the number of discovered genetic signals by 24.7%. The approach can identify novel loci of clinical relevance, support causal gene identification, and reveal complex networks of interacting proteins. Taken together, our study adds significant value to the genetic insights that can be derived from the UKB proteomics data and motivates the wider use of ratios in large-scale GWAS.

## INTRODUCTION

Large-scale studies of the blood circulating proteome leverage the natural variation in the general population to identify genetic and nongenetic factors that control blood protein levels.[1] Of particular interest for drug development are genome-wide association studies (GWASs) that identify protein quantitative trait loci (pQTLs), because they provide genetic evidence for a causal effect of the underlying variant, and hence the affected gene(s), on the levels of the associated protein(s) and their physiological effects. In cases of *cis*-pQTLs, in which the genetic variant is located in proximity of the gene coding for the associated pQTL protein, the effect is most likely through a causal variant that modifies transcription, translation, or stability of the *cis*-encoded protein. More complex, but also more rewarding in terms of potential biological insights, are *trans*-pQTLs because they suggest direct or indirect protein-protein interactions between the presumably causal *cis*-encoded protein and the associated *trans*-protein, which can extend into larger networks when multiple proteins are associated with the same variant and ideally also clinical endpoints of interest.

Such genetics-driven insights are of the highest value to pharmaceutical companies because they can inform drug target discovery and validation, generate hypotheses on modes of action, and suggest biomarkers for target engagement and efficacy. Early successes of pQTL studies[2–5] led to the creation of the UK Biobank Pharma Proteomics Project (UKB PPP) consortium, a precompetitive consortium of 13 biopharmaceutical com-

panies that funded the measurement of over 54,000 UKB samples on the Olink Explore 1536 affinity proteomics platform. Olink uses a dual antibody binding technique, called the proximity extension assay, to quantify the abundance of almost 1,500 blood circulating proteins (Table S1). The UKB PPP consortium recently published initial results from a GWAS that identified over 10,000 pQTLs using this platform.[6] The Olink proteomics data were released in April 2023 to the public and can be accessed and analyzed using the DNAnexus UKB Research Analysis Platform (RAP) platform (ukbiobank.dnanexus.com). The aim in this paper was to explore new methods to enhance pQTL discovery and interpretation using this exceptional and freely available dataset.

My group and others previously developed analysis strategies for GWAS with metabolomics data,[7,8] a field that is similar in many ways to that of pQTL studies. In particular, we showed that partial correlations between metabolites can reconstruct metabolic networks[9,10] and that the hypothesis-free testing of all of the ratios between metabolites can substantially strengthen the association signals, in several cases elevating genetic loci out of the background noise.[11,12] Both approaches are related in that they identify biological relationships between individual molecules through their shared genetic and nongenetic variance, which can then be integrated into larger metabolic networks, such as the atlas of genetic influences of the human metabolome[13] and more recent versions thereof.[14] Previous GWAS with proteomics suggest that Gaussian graphical models (GGMs) built from partial correlations and ratios between protein

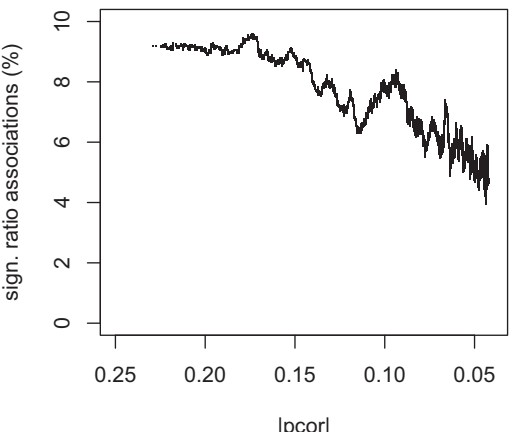

**Figure 1. Percentage of Bonferroni significant protein ratio pair associations (p-gain >10 × 179,923) as a function of the partial correlation |pcor| between the protein pair**

A moving average with a window size of 10,000 data points was used.

levels can reveal biologically relevant protein-protein interactions,[5] but the approach has never been tested at scale.

This article hypothesizes that a GWAS with ratios between protein levels can identify associations and novel links between protein pairs that have not been identified using current GWAS approaches. However, the computational costs of conducting a full-fledged all-against-all ratio GWAS are prohibitive at this point, estimated to several hundred thousand pounds Sterling on the DNAnexus AWS-based platform for a single run of a full-fledged all-against-all ratio GWAS, not considering costs associated with the handling of the generated data. This challenge will be aggravated in the future by the expected increases in proteome coverage.

This article, therefore, takes a more economic approach and tests genetic associations with ratios between proteins that are partially correlated and therefore more likely to be related through some biological process.[15] For each pQTL reported by the UKB PPP consortium that implicated one of two partially correlated proteins, the ratio between the levels of these two proteins is tested for association with the pQTL variant. Then, a GWAS is conducted on those ratios that increased the strength of association at an already known pQTL locus (see the flowchart of this study in Figure S1). It will be shown in the following that by using this approach pQTLs could be identified that were not discovered by the standard GWAS with protein levels conducted by the UKB PPP consortium,[6] and furthermore, that genetic associations with ratios can uncover biologically relevant links between two or more proteins based on their shared genetic and nongenetic variance. Selected cases of biomedical interest are discussed and an interpretation of why I believe ratios work is provided.

## RESULTS

### Identification of ratio QTLs (rQTLs) at established pQTL loci

The increase in the strength of an association with ratios is quantified by the p-gain, which is defined as the smaller of the two

p values for the single-protein associations divided by that for the ratio association.[11] A p-gain of 10 is the equivalent of a nominal p value for a single test; in other words, a p-gain of 10 is expected to be observed by chance in 5% of the cases when ratios between two random proteins are tested. The following requires Bonferroni levels of significance for p values and p-gains throughout and refers to protein ratio associations with significant p-gains as rQTLs.

Quality control (QC) of the Olink data was performed by the UKB PPP before the data was shared with the user. These QC steps include outlier removal and removal of samples of low quality. The number of data points that passed QC was above 97.6% for all Olink panels (see method details). We split the UKB cohort into a discovery set comprising 43,000 individuals and a replication set of 8,700 individuals based on the data field "genetic ethnic grouping" being equal/not equal to "Caucasian" (see UKB documentation on data field 22006) and further limit the analysis to samples collected at baseline.

A total of 179,923 ratio-variant pairs were tested for association, selected as the overlap of 11,936 Bonferroni-significant GGM edges ($p < 4.7 \times 10^{-8}$ or $|pcor| > 1.76 \times 10^{-3}$; Table S2) and 10,248 Bonferroni-significant pQTLs ($p < 3.4 \times 10^{-11}$; Table S3) from the UKB PPP GWAS.[6] A total of 10,760 ratio associations (5.98%) had a Bonferroni-significant p-gain (>10* 179,923), and of these 4,248 (41.4%) replicated in the genetically "non-Caucasian" cohort (p-gain >10 × 10,760). The 4,248 replicated ratio associations covered 2,821 unique protein pairs between 1,001 of the 1,463 (68.4%) proteins assayed on the Olink platform and 926 of the 5,717 (16.2%) genetic variants reported as pQTL variants by the UKB PPP GWAS (Tables S4 and S5). The likelihood of finding a significant ratio association for a protein pair increased with the strength of their partial correlation from ~5% for uncorrelated proteins to 9% for |pcor| ~0.2 (Figure 1), supporting our choice to prioritize GGM protein pairs.

A selection of pharmaceutically relevant rQTLs is provided in Table 1, including associations with a ratio between a *cis*- and a *trans*-located protein, where the *cis*-protein is the target of an approved drug (*Tclin* according to Pharos[16]).

### Identification of rQTLs in a GWAS with ratios

A GWAS was conducted on the 2,821 ratios using the genotyped UKB data. For each ratio, I retained the strongest associations that reached a Bonferroni level of significance of p value $<5 \times 10^{-8}/2,821$, a p-gain $>10^7 \times 2,821$, and that were more distant than 1 million bp from any other significant association with the same ratio. I identified 8,462 ratio-variant pairs with 2,095 unique variants that satisfied this criterion, which corresponds to a discovery per tested ratio of on average three independent GWAS signals with a significant p-gain (Figures 2 and S2; Table S6). The ratios with the largest number of rQTLs discovered were heparin-binding EGF-like growth factor (HBEGF) divided by platelet-derived growth factor subunit A (PDGFA) (N = 25) and integrin subunit β1 binding protein 2 (ITGB1BP2) divided by microtubule interacting and transport, domain containing 1 (MITD1) (N = 24). A total of 999 proteins were implicated in at least one rQTL, with a median of eight rQTLs per protein. The two most frequently occurring proteins were ITGB1BP2, with 259 rQTLs, and ectodysplasin A receptor, with 237 rQTLs. A total of 2,527 (29.9%) of the 8,462 rQTLs were more distant than $10^6$ bp

**Table 1. rQTLs that implicate drug targets in the *cis*-position**

| *cis*-Protein | Ratio | Chr | Pos | rsID | −logP.1 | −logP.2 | −logP.3 | logPgain | GWAS traits |
|---|---|---|---|---|---|---|---|---|---|
| CA6 | CA6/DNER | 1 | 9,034,598 | rs3765963 | 2,878.1 | 0.6 | 3,235.0 | 356.9 | — |
| LEPR | IL6ST/LEPR | 1 | 66,073,982 | rs10399687 | 0.1 | 225.1 | 270.6 | 45.5 | Blood cell traits |
| SLAMF7 | ICAM3/SLAMF7 | 1 | 160,720,074 | rs11581248 | 3.7 | 2,094.1 | 2,631.3 | 537.2 | — |
| NECTIN4 | NBL1/NECTIN4 | 1 | 161,049,509 | rs35434391 | 0.4 | 396.6 | 643.6 | 247.0 | Blood cell traits |
| SELP | SELP/VSIR | 1 | 169,563,951 | rs6136 | 349.1 | 0.3 | 663.3 | 314.2 | Activated partial thromboplastin time |
| DPP4 | DPP4/ITGB1 | 2 | 162,930,725 | rs13015258 | 229.0 | 0.9 | 317.5 | 88.5 | — |
| PDCD1 | PDCD1/TNFRSF8 | 2 | 242,801,752 | – | 271.2 | 0.3 | 364.4 | 93.1 | — |
| CD38 | CD38/RELT | 4 | 15,775,851 | rs28703311 | 894.1 | 0.9 | 1,030.5 | 136.4 | — |
| PDGFRA | IL6ST/PDGFRA | 4 | 55,139,771 | rs35597368 | 0.6 | 364.8 | 439.3 | 74.4 | Impedance of the legs, peak expiratory flow |
| F2R | DAG1/F2R | 5 | 76,028,124 | rs168753 | 0.4 | 56.1 | 276.6 | 220.5 | — |
| FLT4 | FLT4/ICAM2 | 5 | 180,057,293 | rs34221241 | 389.3 | 0.6 | 516.6 | 127.4 | — |
| IGF2R | CTSO/IGF2R | 6 | 160,409,894 | rs75474551 | 106.3 | 366.8 | 434.3 | 67.5 | — |
| AKR1B1 | AKR1B1/SUGT1 | 7 | 134,135,621 | rs2229542 | 92.6 | 0.4 | 172.2 | 79.5 | — |
| IMPA1 | IMPA1/TBCC | 8 | 82,583,771 | rs1967328 | 176.4 | 0.9 | 329.5 | 153.0 | — |
| CA1 | CA1/HMBS | 8 | 86,256,210 | rs12544332 | 35.7 | 0.5 | 114.2 | 78.5 | Blood cell traits |
| CA3 | CA1/CA3 | 8 | 86,351,051 | rs2072696 | 6.7 | 109.1 | 387.6 | 278.6 | — |
| CD274 | CD274/EFNA4 | 9 | 5,453,260 | rs822340 | 446.6 | 0.6 | 500.4 | 53.8 | Ulcerative colitis |
| IL2RA | IL-2RA/TNFRSF4 | 10 | 6,095,928 | rs12722497 | 1,440.8 | 0.1 | 1,892.6 | 451.8 | Streptococcal throat infections |
| LAG3 | LAG3/VCAM1 | 12 | 6,885,076 | rs3782735 | 137.5 | 0.6 | 169.1 | 31.6 | — |
| TXNRD1 | NUDT5/TXNRD1 | 12 | 104,707,047 | rs201402862 | 0.4 | 20.3 | 41.8 | 21.5 | — |
| FLT3 | FLT3/FLT3LG | 13 | 28,637,838 | rs9554228 | 34.2 | 36.0 | 53.9 | 17.8 | Age at menarche, blood cell traits, body size, mass and fat traits, rheumatoid arthritis |
| IL-4R | CKAP4/IL-4R | 16 | 27,327,214 | rs8060025 | 0.3 | 449.7 | 497.6 | 48.0 | Asthma, eosinophils |
| CA5A | AGXT/CA5A | 16 | 87,927,222 | rs55870502 | 0.9 | 2,387.8 | 3,738.1 | 1,350.3 | Basophils |
| COMT | COMT/HNRNPK | 22 | 19,951,271 | rs4680 | 1,008.4 | 0.7 | 3,043.4 | 2,035.1 | Basal metabolic rate, systolic blood pressure, body fat traits |

Selected rQTLs that include a ratio between a *cis*- and a *trans*-located protein, with the *cis*-protein being the target of an approved drug (*Tclin* according to Pharos[16]). The negative $\log_{10}$-transformed p values for the association with the single proteins (−logP.1, −logP.2) and the ratio (−logP.3) and the $\log_{10}$-transformed p-gain (logPgain) for the rQTL are reported; the GWAS traits were annotated using PhenoScanner[17] (details in Table S4).

from any pQTL reported by the UKB PPP GWAS for one of the two proteins in the respective ratio and thus represent previously nonreported pQTLs, which corresponds to an increase of 24.7% in genetic signals derived from the UKB PPP Olink data using ratios compared to the standard approach.

To investigate whether these rQTLs provided new insights of biomedical interest, I annotated the 2,095 rQTL variants identified in this study and the 5,717 pQTL variants reported by the UKB PPP GWAS using PhenoScanner[17] for association with 446 distinct GWAS traits (Tables S6 and S3, respectively). I identified 322 rQTL variants that were more distant than $10^6$ bp from any pQTL variant on the same GWAS trait, implicating 874 rQTLs in a total of 4,700 coassociations with GWAS traits (Table S7).

These rQTLs provide new evidence to support drug target selection. For instance, rs3764640 associated with the ratio STK11/USP8 ($-\log_{10}(p) = 13.8$, $\log_{10}(p\text{-gain}) = 10.5$). The variant is an intragenic SNP in the *STK11* gene and associated with the presence versus absence of psychosis in Alzheimer disease (AD) cases.[18] STK11 is a serine/threonine-protein kinase, and USP8 may play a role in the degradation of activated protein kinases by ubiquitination,[19] which would explain the significant p-gain for the ratio. This rQTL hence not only supports a role of STK11 in AD pathology but it also provides further insights into the putative underlying biological pathways, suggesting that medicinal modification of STK11 or its phosphorylation targets may affect the AD-related phenotypes.

A second example is a region of high linkage disequilibrium (LD) on chromosome 5 (Figure 3), which is a major risk locus for inflammatory bowel disease (IBD).[20] The most likely causal gene prioritized by multiple GWAS, based on its function and the presence of

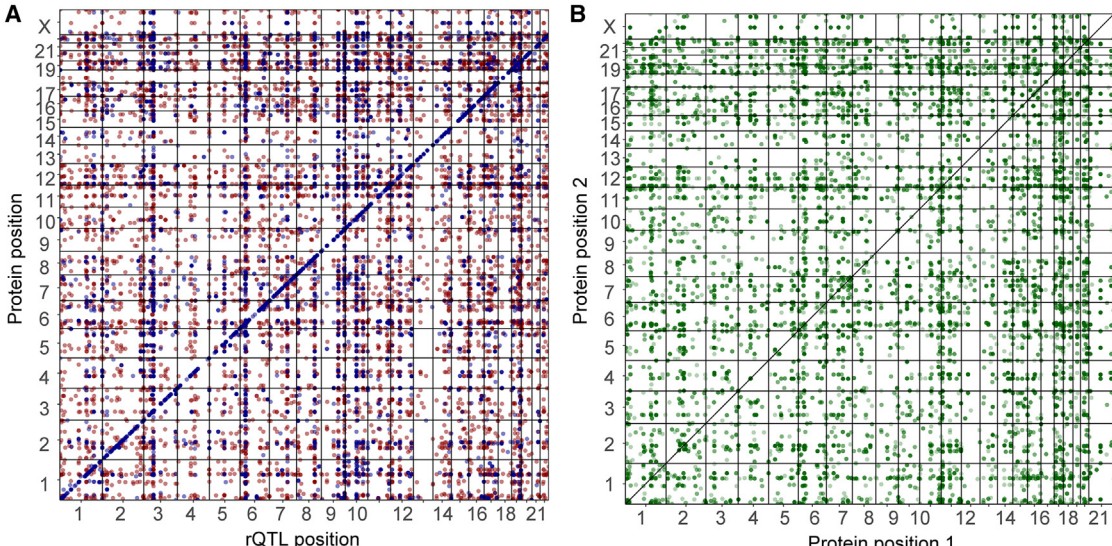

**Figure 2. Two-dimensional Manhattan plots**
(A) The position of the rQTL plotted against the position of the genes coding for the two proteins in the ratio, the stronger of the two single protein associations is in blue, the weaker in red.
(B) The positions of the genes coding for the two proteins in an rQTL ratio plotted against each other; darker colors indicate multiple rQTLs with the same ratio.

an amino acid-changing variant, was *macrophage stimulating 1 (MST1)*. However, this view has been challenged, proposing *glutathione peroxidase 1* (*GPX1*) as a causal gene instead, supported by biochemical experiments showing that a cosegregating amino acid-changing variant in *GPX1* reduced the activity of this antioxidant enzyme.[21] Here, 14 ratios between 16 proteins were identified that associated with a significant p-gain at this locus (Table S8). Seven were ratios of pyruvate kinase, liver, and red blood cells (*PKLR*) with proteins involved in hemoglobin metabolism, including hydroxyacylglutathione hydrolase (HAGH), hydroxymethylbilane synthase (HMBS), arginase 1, and biliverdin reductase B. The biochemical properties of these proteins clearly support a causal role for GPX1 in an oxidative stress-related phenotype, likely related to hemoglobin metabolism in red blood cells. However, four of the ratios were with the *cis*-encoded protein dystroglycan 1 (DAG1) and several proteins not related to red blood cell metabolism, suggesting the presence of a second, likely independent, causal gene at this locus, which would cosegregate with the *GPX1* variant due to the high LD in this region. Whether both pathways are driving factors of the IBD association requires further investigation. Important for the present study is that this case exemplifies the kind of insights that can be drawn from using rQTLs and their value for drug target evaluation and hypothesis generation.

### Discovery of *cis*-pQTLs

Observation of *cis*-pQTLs is considered genetic evidence to confirm the target specificity of the respective affinity binding assay. Sun et al.[6] found a *cis*-pQTL for 1,163 (79.5%) of the 1,463 assayed proteins. Here, I report 39 additional genetic variants that associated with a ratio that involves a protein located in-*cis* and a second protein located in-*trans* (Table 2). These *cis*-pQTLs became presumably discoverable because the

*trans*-proteins in the ratios captured some unidentified shared nongenetic variance, accounting for which led to the significant p-gains. The corresponding 39 proteins include three Olink targets for which no genetic signal had been found in the UKB PPP GWAS at all (ARHGEF12, EIF4EBP1, INPPL1) and thus provide genetic evidence that the respective antibodies bind their designated targets. Even by using only a subset of all of the possible ratios, I identified 13% of the 300 *cis*-pQTLs that were not accounted for so far, increasing confidence in the target specificity of the Olink platform for these proteins. More may be identified in an all-against-all ratio approach.

### Refinement of the rQTL loci

For economic reasons, the GWAS was conducted using the genotyped variants only and therefore I may have missed variants of interest. For each of the 8,462 rQTLs, I therefore refined the associations within ±500,000 bp of the respective lead variant by using the imputed UKB genotype data, both in the discovery and in the replication cohort. The summary statistics are provided for all 8,462 refined regions on FigShare (Data S1, https://doi.org/10.6084/m9.figshare.23695398). These data can also be used to further refine the loci of interest—for instance, to identify potentially multiple independent signals using SuSiE[23] or to test for colocalization with other traits of interest using coloc.[24] To visualize individual rQTLs regional association plots were generated for all rQTLs, both in the discovery and the replication cohort (Figure S3).

I then used coloc[24] to ask whether the two proteins in a ratio shared the same genetic signal (Q.12), whether any of the two proteins shared a signal with the ratio (Q.13 and Q.23), and whether the signal for the ratio was shared between discovery and the replication cohort (Q.33.repli). Table S6 provides the most likely hypothesis for each of these four questions, together with its posterior probability. In 7,414 (87.6%) of the 8,462 cases, at least one of

## HAGH/PKLR

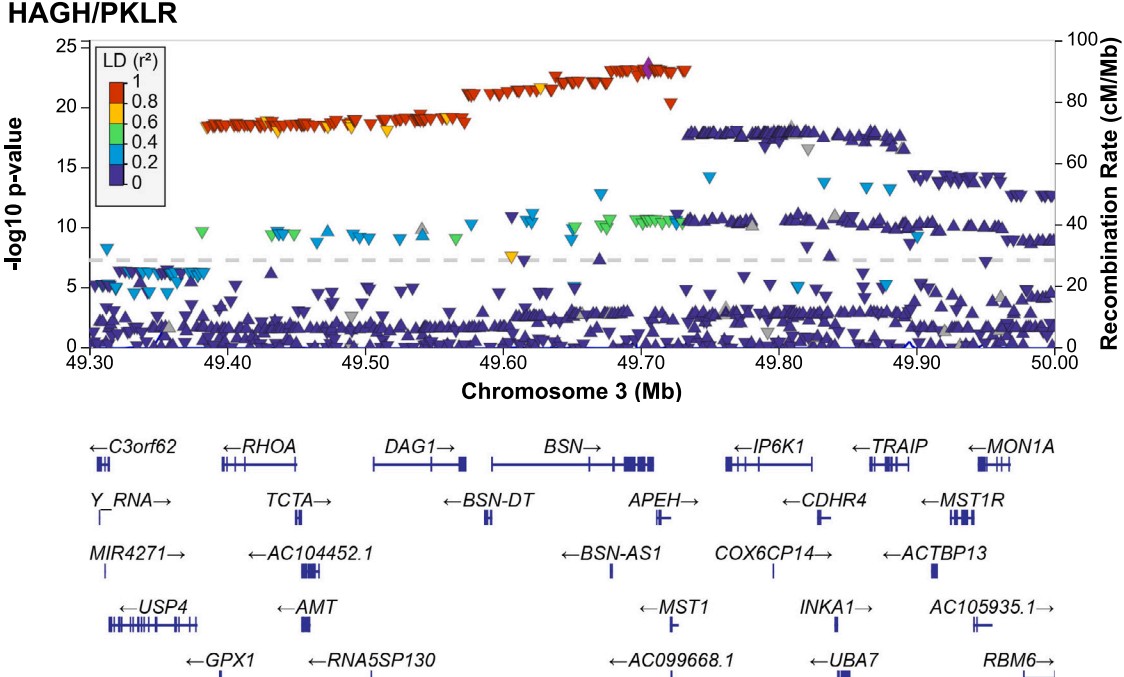

**Figure 3. Regional association plot for the association of the HAGH/PKLR ratio at a major IBD locus on Chr3**
Plot created using LocusZoom.[22]

the proteins shared a genetic signal with the ratio (Q.13 = H4 or Q.23 = H4), in 1,305 (15.4%) cases both proteins shared a signal with the ratio (Q.13 = H4 and Q.23 = H4), and in 489 (5.8%) cases there was no signal detectable for either of the two proteins alone (Q.13 = H2 and Q.23 = H2 and Q.12 = H0). A total of 6,775 of the 8,462 rQTLs (80.1%) shared a genetic signal between discovery and the replication cohort (Q.33.repli = H4).

For each rQTL region, the variant with the strongest association with the ratio in the discovery cohort was designated as the lead variant, and I asked whether the association on this variant replicated. Requiring in the discovery cohort $p < 5 \times 10^{-8}/2{,}821$ and p-gain $>10 \times 10^{6} \times 2{,}821$ and in the replication cohort $p < 0.05/8{,}462$ and p-gain $>10 \times 8{,}462$, I identified 4,181 rQTLs (49.4%) that satisfied this stringent Bonferroni significance criterion. Considering that 80.1% of the rQTLs shared the same genetic signal between discovery and the replication cohort, it is likely that more rQTLs can be replicated when more samples become available.

### Why do ratios work and what do they represent?
With $P_1$ and $P_2$ representing the levels of two blood circulating proteins (the indices of the individual samples were suppressed) two linear models can be fit to the log-scaled protein levels by selecting parameters $\alpha_i$, $\beta_i$, and $\gamma_i$ such that they minimize the square of the nonexplained variance $\varepsilon_i$ in the following equation:

$$\log(P_i) = \alpha_i + \beta_i \times \text{SNP} + \gamma_i \times W + \varepsilon_i \text{ for } i \in \{1, 2\}$$

SNP represents the number of effect alleles (0, 1, 2) of a given genetic variant in a given sample and $W$ denotes some noni-

dentified nongenetic variance that is shared by both proteins. Using the identity $\log(A/B) = \log(A) - \log(B)$, the ratio can then be written as:

$$\log(P_1 / P_2) = (\alpha_1 - \alpha_2) + (\beta_1 - \beta_2) \times \text{SNP}$$
$$+ (\gamma_1 - \gamma_2) \times W + (\varepsilon_1 - \varepsilon_2).$$

Because the significance level (p-value) of the association with the variant depends on the proportion of the variance that is explained by the genetic term (SNP) compared to the remaining variance ($W + \varepsilon$), the strength of the association with the ratio can increase under two conditions: (1) when $\beta_1$ and $\beta_2$ have opposite signs, or (2) when $\gamma_1$ and $\gamma_2$ are of comparable size and nonzero.

$\beta_1$ and $\beta_2$ having opposite signs implies that the genetic variant increases the levels of one protein while decreasing those of the other (Figure 4). When working with metabolites, this situation can occur when the ratio represents a substrate-product pair of an enzyme whose efficacy is affected by the genetic variant. Many such cases have been reported.[12,13,25] For proteins, a possible scenario is a genetic variant that increases the expression of one protein that acts as a suppressor of a second protein. One example is the association of rs1065853, which is in LD with coding SNP rs7412 in *APOE*, and the ratio between low-density lipoprotein receptor (LDLR) and proprotein convertase subtilisin/kexin type-9 (PCSK9) protein levels ($\log_{10}$(p-gain) = 67.2). PCSK9 binds LDLR and targets it for degradation.[26] The availability of PCSK9 to degrade LDLR in turn is limited by binding to apolipoprotein B,[27] the levels of which are associated with rs7412 in *APOE*.

**Table 2. Novel *cis*-pQTLs**

| *cis*-Protein | Ratio | Chr | Pos | rsID | −logP.1 | −logP.2 | −logP.3 | logPgain |
|---|---|---|---|---|---|---|---|---|
| MNDA | MNDA/NCF2 | 1 | 158,788,542 | rs2875712 | 5.7 | 0.3 | 16.9 | 11.2 |
| EPCAM | EPCAM/GPA33 | 2 | 47,773,540 | rs6708696 | 1.6 | 1.3 | 15.5 | 13.9 |
| ANXA4 | ANXA4/LACTB2 | 2 | 70,033,584 | rs2228203 | 11.6 | 3.8 | 232.7 | 221.1 |
| TMSB10 | DBI/TMSB10 | 2 | 85,133,861 | rs13409738 | 0.0 | 8.1 | 36.8 | 28.7 |
| NCK2 | NCK2/PLA2G4A | 2 | 106,452,253 | rs10169998 | 4.6 | 0.0 | 36.4 | 31.8 |
| TGFBR2 | TGFBR2/TNFRSF1A | 3 | 30,729,510 | rs114836705 | 8.2 | 0.4 | 25.0 | 16.9 |
| PPP1R2 | PPP1R2/SNAP23 | 3 | 195,238,559 | rs34950021 | 1.9 | 0.8 | 26.6 | 24.7 |
| CXCL3 | CXCL3/CXCL5 | 4 | 74,797,139 | rs352024 | 0.6 | 514.2 | 1,229.3 | 715.1 |
| FYB1 | FYB1/PPP1R12A | 5 | 39,338,358 | rs3822462 | 1.8 | 0.7 | 14.1 | 12.3 |
| DAB2 | DAB2/NCK2 | 5 | 39,427,481 | rs75839063 | 2.5 | 0.0 | 20.3 | 17.8 |
| HBEGF | HBEGF/PDGFA | 5 | 139,720,400 | rs2237077 | 7.6 | 0.8 | 31.1 | 23.4 |
| PDLIM7 | PDLIM7/SRC | 5 | 176,922,643 | rs335428 | 7.5 | 1.0 | 28.9 | 21.4 |
| MPIG6B | MPIG6B/PLXNA4 | 6 | 31,346,193 | rs2507982 | 0.3 | 5.1 | 18.5 | 13.5 |
| MAP3K5 | MAP3K5/MAVS | 6 | 136,888,889 | rs56379668 | 4.0 | 1.9 | 31.7 | 27.7 |
| VTA1 | RWDD1/VTA1 | 6 | 142,641,606 | rs12189801 | 0.6 | 4.2 | 21.1 | 16.9 |
| LAT2 | CLIP2/LAT2 | 7 | 73,780,812 | rs512023 | 4.9 | 4.8 | 59.6 | 54.7 |
| CASP2 | CASP2/NUB1 | 7 | 142,986,684 | rs3181165 | 2.8 | 0.5 | 14.3 | 11.4 |
| PLPBP | PLPBP/RWDD1 | 8 | 37,635,649 | rs7463174 | 8.7 | 1.1 | 37.9 | 29.2 |
| EIF4EBP1[a] | DNPH1/EIF4EBP1 | 8 | 37,884,310 | rs28797500 | 0.5 | 4.3 | 18.4 | 14.1 |
| LYN | KIFBP/LYN | 8 | 56,785,133 | rs6985703 | 0.2 | 5.5 | 18.5 | 12.9 |
| INPPL1[a] | BANK1/INPPL1 | 11 | 72,064,041 | rs79658353 | 0.1 | 1.7 | 13.0 | 11.3 |
| PPME1 | ATG4A/PPME1 | 11 | 73,948,875 | rs79153613 | 0.6 | 9.5 | 30.1 | 20.6 |
| ARHGEF12[a] | ARHGEF12/AXIN1 | 11 | 120,278,477 | rs34172482 | 4.6 | 0.1 | 37.7 | 33.1 |
| IRAG2 | CRACR2A/IRAG2 | 12 | 25,243,115 | rs1908946 | 0.3 | 3.5 | 16.4 | 12.8 |
| METAP2 | EIF4B/METAP2 | 12 | 95,830,338 | rs159853 | 0.6 | 8.9 | 27.4 | 18.5 |
| TRIAP1 | TMSB10/TRIAP1 | 12 | 120,902,007 | rs542407 | 0.3 | 3.6 | 14.7 | 11.1 |
| SRP14 | APEX1/SRP14 | 15 | 40,325,829 | rs6492926 | 0.0 | 10.9 | 25.3 | 14.4 |
| SNAP23 | CD69/SNAP23 | 15 | 42,808,309 | rs73404730 | 0.8 | 1.7 | 21.2 | 19.6 |
| PPIB | MANF/PPIB | 15 | 64,780,971 | rs73452261 | 1.0 | 1.9 | 29.2 | 27.3 |
| MESD | MANF/MESD | 15 | 81,279,706 | rs57967327 | 0.3 | 5.4 | 74.5 | 69.1 |
| CORO1A | CORO1A/TBC1D5 | 16 | 30,147,265 | rs7201780 | 4.9 | 0.0 | 18.2 | 13.3 |
| STX4 | DNMBP/STX4 | 16 | 31,004,812 | rs12445568 | 0.3 | 6.7 | 20.5 | 13.9 |
| AHSP | AHSP/CA2 | 16 | 31,463,216 | rs4889659 | 9.9 | 0.0 | 26.3 | 16.4 |
| VPS53 | SCAMP3/VPS53 | 17 | 400,933 | rs9916346 | 0.3 | 2.9 | 14.7 | 11.8 |
| STK11 | STK11/USP8 | 19 | 1,207,238 | rs3764640 | 3.3 | 0.6 | 13.8 | 10.5 |
| CDKN2D | CDKN2D/TACC3 | 19 | 9,868,278 | rs10420364 | 0.5 | 6.7 | 45.5 | 38.7 |
| CDC37 | CDC37/PLA2G4A | 19 | 10,523,086 | rs10854116 | 4.4 | 0.8 | 47.0 | 42.6 |
| TBCB | MITD1/TBCB | 19 | 36,593,915 | rs61741470 | 0.3 | 9.7 | 130.7 | 121.0 |
| CRKL | CRKL/DBNL | 22 | 21,139,239 | rs117858197 | 1.6 | 0.5 | 34.8 | 33.1 |

List of 39 genetic variants that associated with a ratio that involves a protein located less than 1 MB from the variant (*cis*-protein) and that has no *cis*-pQTL in the UKB PPP GWAS. The negative $\log_{10}$-transformed p values for the association with the single proteins (−logP.1, −logP.2) and the ratio (−logP.3) and the $\log_{10}$-transformed p-gain (logPgain) for the rQTL are reported (details in Table S6).
[a]This protein has no pQTL in the UKB PPP GWAS.

If only one of the proteins is affected by the genetic variant, then observing a significant p-gain implies that $\gamma_1$ and $\gamma_2$ must be of comparable size and nonzero, and the association with the ratio indicates the presence of some nongenetic variance that is shared by both proteins. For instance, ITGB1BP2 has only five *trans*-pQTLs in the UKB PPP GWAS of moderate effect size, but occurs with 26 different ratios in 259 rQTLs in the present study's GWAS, the strongest with a log10(p-gain) = 1,115.5 for the association of rs4680 with the ratio catechol-*O*-methyltransferase (COMT) divided by ITGB1BP2 at the *COMT* gene locus. The association of ITGB1BP2 with rs4680 was not significant (p = 0.69). The ratio with the largest number of rQTLs

| | Shared genetic variance | Shared non-genetic variance |
|---|---|---|
| *cis*-rQTL | | |
| *trans*-rQTL | | |

**Figure 4. Possible scenarios that can lead to a significant p-gain in a ratio association**
P1 and P2 are the proteins in the ratio that associates with the genetic variant *SNP*, X is the causal *cis*-encoded protein in the case of a *trans*-rQTLs, and W denotes some unidentified shared nongenetic variance.

in the GWAS was ITGB1BP2/MITD1, which had 24 rQTLs compared to only 2 pQTLs for MITD1 in the UKB PPP GWAS (Figure 5). Intriguingly, both proteins are highly correlated (Pearson $r^2 = 0.86$), suggesting that their correlation is driven by some shared but not identified factor. The strongest correlations with one of the clinical biochemistry and blood traits available in UKB was with platelet count ($r^2 = 0.12$ with ITGB1BP2, $r^2 = 0.10$ with MITD1, and $r^2 = 0.025$ with the ratio), which are too weak to explain the full correlation between both proteins, suggesting that a driving factor for this association is related to some more specific, probably blood cell-type-related trait that is not readily available in the UKB phenotype dataset.

There are multiple possible causes that can lead to a significant p-gain in a ratio association, as schematized in Figure 4; some rQTLs reveal the presence of shared genetic variance, whereas others suggest the proteins in the ratio being linked through some shared nongenetic processes. It is interesting to note in this context that adjusting for hidden factors by using genomic or proteomic surrogates can also lead to a substantial improvement in pQTL detection.[28]

## What can be learned from rQTLs?

To evaluate the enrichment in protein pairs that were linked through significant ratio associations and/or GGM edges, the STRING database of protein-protein interactions was used (Table S5). Of the 2,281 protein pairs, 168 pairs (6.0%) had a protein-protein interaction link in the STRING database with a high confidence score (>0.7), and random pairs between these proteins had on average only 22.1 links (SD = 4.2, based on 100 samplings), which corresponds to a 7.6-fold enrichment. For comparison, of the 11,936 protein pairs linked through significant GGM edges, 465 pairs (3.9%) had a protein-protein interaction reported in STRING, whereas random sampling yielded an average of 89.4 (SD = 9.4), which corresponds to a 5.2-fold enrichment for proteins linked through GGM edges alone.

Using cytokine and cytokine-receptor annotations from CytokineLink,[29] 39 protein pairs were identified with an rQTL that involved cytokine-cytokine pairs, 15 receptor-receptor pairs, and 9 cytokine-receptor pairs, 7 of which were known

and 2 were new (colony-stimulating factor 1:lymphotoxin β receptor and C-X-C motif chemokine ligand 9 [CXCL9]:tumor necrosis factor receptor superfamily member 9 [TNFRSF9]). CytokineLink predicted 1,542 cytokine-cytokine interactions between 77 cytokines from the Olink platform (out of 77 × 76/2 = 2,926 possible interactions). The number of cytokine pairs that were involved in rQTLs was significantly enriched (30 out of 39 compared to 1,512 out of 2,887 with no rQTL, p < 0.002, Fisher exact test).

When multiple proteins associate in rQTLs with the same variant, networks of related proteins can be constructed. Here, an example is presented of how a single pQTL around a pharmaceutically interesting protein can be extended into a network of potentially interacting proteins. NFATC1 (nuclear factor of activated T cells 1) is a key transcription factor and regulator of the immune response[30] and a molecular target for immunosuppressive drugs such as cyclosporin A.[31] NFATC1 has been implicated in the pathogenesis and targeted therapy of hematological malignancies.[32,33] rs657693 is a *cis*-pQTL for NFATC1 in the UKB PPP GWAS and one of only two genetic associations for this protein. rs657693 is identified here as an rQTL for the ratio of NFATC1 with 16 other proteins (AXIN1, BACH1, BANK1, BCR, CASP2, CD69, EIF4G1, FADD, FOXO1, IKBKG, INPPL1, IRAK1, LBR, PTPN6, SPRY2, TJAP1; Table S4), none of which had a significant association with this variant alone. Of the 16 proteins 9 had a second replicated rQTL in a ratio with NFATC1 elsewhere in the genome, and in all of these cases, the other protein in the ratio was the driving pQTL, with 5 of them being *cis*-pQTLs (AXIN1, BANK1, FOXO1, SPRY2, TJAP1). GWAS identified additional rQTLs, including *cis*-rQTLs for BCR (rs713617), cluster of differentiation 69 (CD69) (rs7309767), and *Fas*-associated protein with death domain (FADD) (rs7939734). Ingenuity Pathway Analysis identified a number of functional links between these proteins, generating a network of proteins that can now be linked through genetic evidence and rQTLs to the NFATC1 locus, potentially supporting the development of new immunosuppressive drugs (Figure S4). For instance, it has been shown that silencing NFATC1 results in the phosphorylation of forkhead box O1 (FOXO1) and thereby plays a role in cell differentiation.[34] Expression of NFATC1 and nine of the proteins in its ratio (AXIN1, BCR, CASP2, CD69, EIF4G1, FADD, IRAK1, LBR, PTPN6) is enriched in leukemia cells (false discovery rate = 0.003).[35] Interleukin 1 receptor associated kinase 1 (IRAK1) is an emerging therapeutic target in hematologic malignancies, and it has been suggested that IRAKs participate in regulatory interactions with FADD.[36] Indeed, FADD has

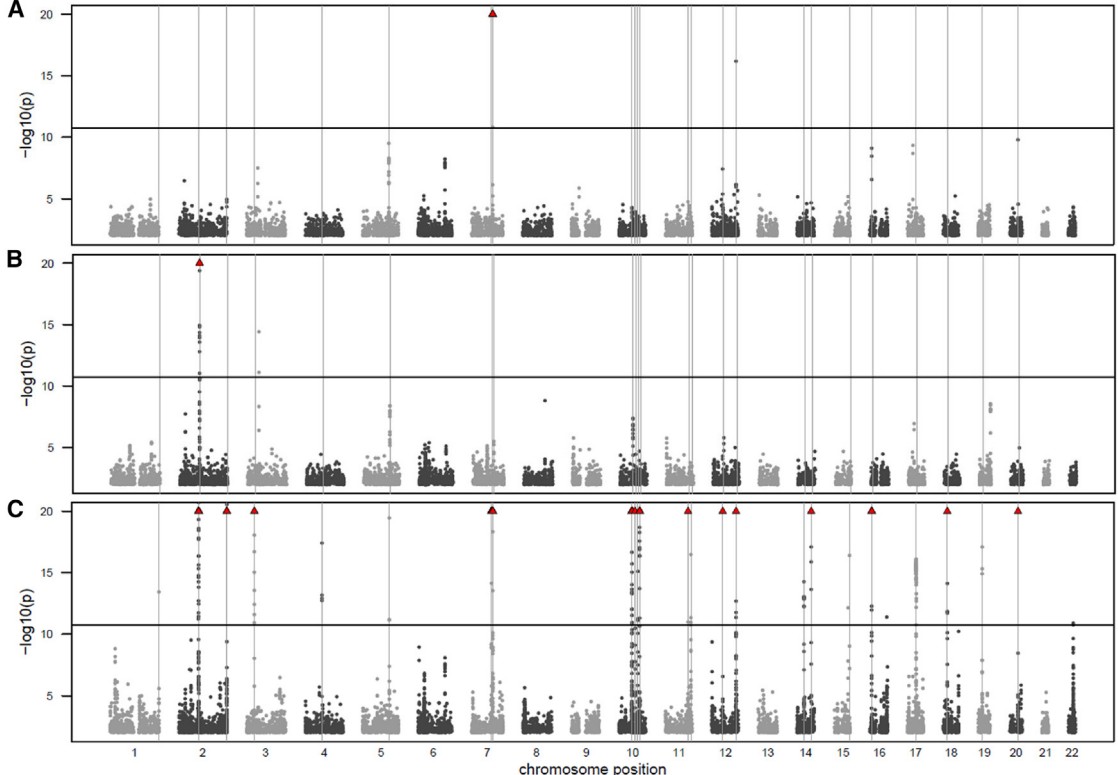

**Figure 5. Example of a ratio that leads to the discovery of novel signals**
(A–C) Manhattan plots for the GWAS with (A) ITGB1BP2, (B) MITD1, and (C) the ratio ITGB1BP2/MITD1. Associations with p values exceeding $10^{-20}$ are indicated by red triangles. Vertical lines indicate 24 Bonferroni-significant (p < 5 × $10^{-8}$/2821) rQTLs for the ratio. Manhattan plots for all 2,821 ratio GWAS are available as online (Figure S1 and Data S1, https://doi.org/10.6084/m9.figshare.23695398).

been shown to physically interact with IRAK1,[37] lending further experimental support to this rQTL-derived network and its role in T cell development.

To further explore the benefits of all-against-all ratios, the associations of SNP rs12075 were computed with all possible ratios between 76 cytokines that are in the Olink panel. rs12075 (1:159175354:G:A) is an amino acid changing variant (c.125G>A, p.Gly42Asp) in the *atypical chemokine receptor 1* gene (*ACKR1* aka *DARC*). The glycine variant defines the Fy[a] allele and the aspartate variant defines the Fy[b] allele of the Duffy blood group system.[38] DARC is clinically important because it is the entry point for the human malaria parasite *Plasmodium vivax*. Individuals with two copies of the Fy[a] allele or a silenced Fy[b] allele are resistant to *P. vivax* infection. A structural basis for DARC binding to *P. vivax*'s Duffy-binding protein involving the region around the p.Gly42Asp variant has been proposed.[39]

Eleven cytokines are associated in the UKB PPP GWAS at the *ACKR1* locus (CCL2 [C-C motif chemokine ligand 2], CCL7, CCL8, CCL11, CCL13, CCL14, CCL17, CCL26, CXCL1, CXCL6, CXCL8; Table S3). CCL7 and CCL8 protein levels increased with the copy number of the Fy[a] allele, whereas levels of the other nine cytokines decreased with that variant. DARC controls chemokine levels through promiscuous binding.[40] The associations with these cytokines thus match the function of DARC. In the discovery study, rs12075 associated with three ra-

tios (CCL13/CCL8, CCL2/CCL7, and CCL11/CCL7; Table S4), and the strongest association of rs12075 was with the ratio between CCL8 and CCL13. An interaction between these two cytokines has been experimentally established by bidirectional immunoligand blotting.[41] Testing the association of rs12075 with all possible ratios between the 76 cytokines on the Olink panel implicated 12 additional cytokines (CCL3, CCL4, CCL14, CXCL11, CXCL12, HGF, IL-7, PDGFA, TGFB1, THPO, TNFSF13, TNFSF14) in significant (p-gain > $10^{10}$) rQTLs (Figure S5; Table S9). It goes beyond the scope of the present study to interpret these associations in further detail. The takeaway message here is that using ratios not only identified additional cytokines that associate with the Duffy blood type but they also suggest interactions between specific pairs of proteins, such as CXCL6, which occurs in a significant ratio with CCL8, but not with CCL7. Especially at pleiotropic loci, where multiple proteins associate with a clinically relevant variant, it may be worthwhile to conduct this kind of all-against-all ratio analysis, using a subset of functionally related protein, as done in this example, or even extending the ratio analysis to the full protein panel.

## DISCUSSION

In this study, a GWAS at scale is presented with ratios between blood circulating protein levels, using the recently released Olink

proteomics data for almost 1,500 proteins measured in the blood plasma of 54,000 participants of the UKB. Using ratios, increases in the strengths of association by up to several hundred orders of magnitude were observed, involving two-thirds of the proteins targeted by the Olink platform, increasing the strength of association at 16% of the pQTLs from the UKB PPP GWAS. Novel *cis*-pQTLs were reported for 13% of the 300 Olink proteins for which such a target-confirmatory QTL had not been identified so far and uncovered over 2,500 novel QTLs with ratios at loci that had not been highlighted by the UKB PPP GWAS using single protein levels, which corresponds to a 25% increase in the discovery rate.

I argue that ratios can account for unidentified genetic and/or nongenetic variance that is shared between the associated protein pairs (Figure 4). rQTL protein pairs were 7.2-fold enriched in known protein-protein interactions, demonstrating that they add substantial new information to hypothesis generation and providing a broad set of protein-protein relationships that can be mined using network pharmacology (the NFATC1 example)[42] and systems immunology (the ACKR1 example)[43,44] approaches. I further reported selected examples that illustrate insights gained from using ratios, adding new information to established loci (the GPX1 with IBD example) and identifying entirely novel loci (the STK11 with AD example).

Because I could only discuss a fraction of the biologically relevant findings in this paper, I freely share the full summary statistics of the GWAS using array-genotyped data and of the refinement using imputed genotype data, together with the corresponding Manhattan and regional association plots. These data represent a rich resource for biomedical hypothesis generation that complements the data generated by the UKB PPP GWAS and should be of particular value for pharmaceutical drug target development.

I hope to have demonstrated the benefits of analyzing ratios between protein levels at scale, an approach that I believe has already shown its benefits in the metabolomics field. Future work is needed to further speed up the calculation of ratio associations, especially in light of the broadening proteomics panels and their increased application in large-scale cohorts. Also, theoretical development and generalization of the concept of ratios to more general classes of multivariate association tests may further improve the power for the detection of genetic signals.

### Limitations of the study

The following caveats apply: I analyzed only a subset of all of the possible ratios. Although the likelihood of finding a significant ratio association for a protein pair increased with the strength of their partial correlation, even for uncorrelated proteins, this likelihood remains high (5%), supporting the testing of all possible ratios in future GWAS, if resources permit. Indeed, I hope that the present study motivates further methods development that could render all-against-all ratio testing computationally feasible, and perhaps also more research into formal statistical methods that may generalize the analysis of combinations of quantitative traits as dependent variables in GWAS.

As for all GWAS, this study is hypothesis generating by nature. An rQTL suggests a potential relationship between the two pro-

teins in the ratios, plus potentially a third protein encoded at the genetic locus in the case of *trans*-rQTLs. These relationships can be of different natures, ranging from a direct physical interaction over indirect regulatory interactions to broad shared nongenetic factors. Also, proteins may have multiple functions and can be associated with multiple pathways, which needs to be considered in the interpretation of the rQTLs on a one-by-one basis. I also acknowledge that there is presently no established causal inference framework, which somewhat limits the interpretation of the findings from a Mendelian randomization perspective.

It should also be noted that affinity proteomics technologies have numerous limitations, such as effects of epitope-changing variants, nonspecific binding, and uncertainty about target specificity. However, the many biologically relevant associations that have been derived using data from the Olink and other affinity proteomics platforms suggest that these concerns are of minor relevance. The fact that 39 novel *cis*-pQTLs were identified in this study provides further confirmatory evidence for the target specificity of their respective affinity binders.

## STAR★METHODS

Detailed methods are provided in the online version of this paper and include the following:

- KEY RESOURCES TABLE
- RESOURCE AVAILABILITY
  - Lead contact
  - Materials availability
  - Data and code availability
- EXPERIMENTAL MODEL AND SUBJECT DETAILS
- METHOD DETAILS
- QUANTIFICATION AND STATISTICAL ANALYSIS
- ADDITIONAL RESOURCES

### SUPPLEMENTAL INFORMATION

### ACKNOWLEDGMENTS

I thank all UKB participants for their contribution. K.S. is supported by the Biomedical Research Program at Weill Cornell Medicine in Qatar, a program funded by the Qatar Foundation. K.S. is also supported by Qatar National Research Fund (QNRF) grant NPRP11C-0115-180010. Graphical abstract created with BioRender.com. The statements made herein are solely the responsibility of the author.

### AUTHOR CONTRIBUTIONS

K.S. conceived the study, conducted the data analyses, and wrote the paper.

### DECLARATION OF INTERESTS

The author declares no competing interests.

**Cell Genomics**
**Article**

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

## STAR★METHODS

### KEY RESOURCES TABLE

| REAGENT or RESOURCE | SOURCE | IDENTIFIER |
|---|---|---|
| **Deposited data** | | |
| pQTL summary statistics from the UKB PPP project (Table S6 of Sun et al.[6]) | Sun et al.[6] | https://doi.org/10.1101/2022.06.17.496443 |
| Genomics, proteomics, and phenotype data of UK Biobank participants | UK Biobank | https://biobank.ndph.ox.ac.uk/showcase/ |
| Summary statistics using imputed UKB genotype data for the regions (+/−500kb) around the 8,462 rQTLs discovered in the GWAS | This study | https://doi.org/10.6084/m9.figshare.23695398 |
| Manhattan plots for the GWAS (PDF format) | This study | https://doi.org/10.6084/m9.figshare.23695398 |
| Regional association plots for the local refinements (PDF format) | This study | https://doi.org/10.6084/m9.figshare.23695398 |
| **Software and algorithms** | | |
| R | The R Project for Statistical Computing | https://www.r-project.org |
| Rstudio | Posit | https://posit.co |
| Genomics analysis tools | DNAnexus | https://ukbiobank.dnanexus.com/landing |
| GeneNet | Schäfer and Strimmer[45] | N/A |
| **Other** | | |
| STRING | Szklarczyk et al.[35] | https://string-db.org |
| NCBI Pharos | Sheils et al.[16] | https://pharos.nih.gov |
| PhenoScanner | Staley et al.[17] | http://www.phenoscanner.medschl.cam.ac.uk |
| CytokineLink | Olbei et al.[29] | https://github.com/korcsmarosgroup/CytokineLink |
| LocusZoom | Boughton et al.[22] | http://locuszoom.sph.umich.edu |
| GeneCards | Stelzer et al.[46] | https://www.genecards.org |
| Plink | Purcell et al.[47] | https://www.cog-genomics.org/plink/2.0 |

### RESOURCE AVAILABILITY

#### Lead contact
Further information and requests for resources and reagents should be directed to and will be fulfilled by the lead contact, Karsten Suhre (kas2049@qatar-med.cornell.edu).

#### Materials availability
This study did not generate new unique reagents.

#### Data and code availability
- All analyzed data was from UK Biobank and obtained through the UKB RAP system under application reference number 43418 and is accessible upon application online via https://biobank.ndph.ox.ac.uk/showcase/.
- Full summary statistics for all 2,821 GWAS with ratios using the array-genotyped UKB data shall be made available via the GWAS catalog.
- Summary statistics using imputed UKB genotype data for the regions (+/−500kb) around the 8,462 rQTLs discovered in the GWAS are available on FigShare (https://doi.org/10.6084/m9.figshare.23695398), both for the discovery and the replication cohort.
- Manhattan and regional association plots based on this data are available in PDF format on FigShare at the same URL.
- This study did not generate any unique code.

## EXPERIMENTAL MODEL AND SUBJECT DETAILS

All data and samples were collected by UK Biobank following all relevant ethical guidelines and procedures and were shared with the UKB users under rules reviewed by the UK Biobank ethics committee and board.

All data was obtained through the UKB RAP system on the DNAnexus platform (data dispensed on April 23, 2023; application id 43418) for samples satisfying the criterion "Number of proteins measured | Instance 0" is greater than "0" (https://biobank.ndph.ox. ac.uk/showcase/field.cgi?id=30900). Imputed genotypes[48] were extracted from BGEN files (https://biobank.ndph.ox.ac.uk/ showcase/label.cgi?id=100319) using bgenix[49] and reformatted to text format (.raw) using plink[47] for further analysis in R. Phenotype data for age, sex, BMI, the first three genotype principal components, and the classification of genetic ethnic grouping were extracted using the DNAnexus cohort browser and the table downloader app (https://ukbiobank.dnanexus.com/landing). Details on genomics and proteomics data QC and preprocessing are available in the accompanying UK Biobank resource files (available at the respective showcase links given above).

The downloaded proteomics dataset comprised NPX values for 1,463 proteins for 52,749 participants. NPX values correspond to relative protein concentrations and are reported on a log-scale. Data analysis was restricted to 52,705 samples that were collected at baseline (instance 0). Samples were split into a discovery set of 43,509 samples identified as Caucasian based on the genetic ethnic grouping variable and a replication set of 9,196 ethnically diverse samples (https://biobank.ndph.ox.ac.uk/showcase/field.cgi? id=22006). A total of 5,717 unique variants corresponding to 10,248 pQTLs of the UKB PPP GWAS were analyzed. These pQTLs were obtained from Table S6 of Sun et al.[6]

## METHOD DETAILS

Partial correlations were computed using the R function ggm.estimate.pcor from the package GeneNet.[45] All baseline samples were used for this step. As this analysis does not allow for the presence of missing values, samples with more than 20% missing protein values were removed (N = 1,840), followed by proteins that were missing in more than 20% of the samples (N = 3). The remaining missing data points were imputed to minimum (N = 37,419). A total of 11,936 partial correlations were identified at a Bonferroni significance cut-off p value of $4.7 \times 10^{-8}$. The smallest $r^2$ at this level was 0.00176.

The GWAS on 2,821 ratios was conducted using plink2[47] on the UKB RAP platform hosted by DNAnexus with the –glm option, using age, sex, and the first the genoPCs as covariates. We used the array-genotyped UKB data with the following variant filtering options: –geno 0.1, –hwe 1e−15, –mac 100, –maf 0.01, –mind 0.1.

## QUANTIFICATION AND STATISTICAL ANALYSIS

Linear models with inverse-normal scaled proteomics data (NPX values) as dependent variables and genotype, age, sex, and the first three genotype principal components were computed using the R function "lm". For ratios, inverse-normal scaled differences between the two NPX values were used, based on the relation log(A/B) = log(A) - log(B) and the NPX values representing protein levels on a log-scale. The p-gain for associations with ratios between two protein traits was computed as the smaller of the two p values for the individual trait associations divided by the p value for the ratio association.[11] Log10-scaled p values and p-gains were used throughout to avoid numeric overflows and rounding of small p values to zero.

For all pairs of proteins with a Bonferroni significant partial correlation all variants that were associated with at least one of the two proteins in a pQTL in the UKB PPP GWAS were identified. For these variant – protein pairs the single protein and ratio association statistics were computed using the discovery and replication samples separately.

## ADDITIONAL RESOURCES

The STRING database of proteins and their functional interactions was used to identify known relationships between proteins.[35] The database was downloaded from https://string-db.org/cgi/download (version 11.5, accessed 5 May 2023). Annotated cytokine and cytokine-receptor pairs were downloaded from CytokineLink[29] (https://github.com/korcsmarosgroup/CytokineLink, accessed 30 June 2023). Drug target development status was obtained from NCBI Pharos[16] (accessed 11 July 2023). Variants were annotated with PhenoScanner API[17] using proxies based on EUR LD $r^2 > 0.8$ (accessed 5 May - 11 July 2023). LocusZoom[22] was used to generate regional association plots with LD annotation (EUR population). GeneCards[46] was used to obtain general information about the associated genes and proteins.

