## [Document S2. Transparent peer review records for Karsten Suhre · Cell Genomics]

Genetic associations with ratios between protein levels detect new pQTLs and reveal protein-protein interactions.

Karsten Suhre^{1,2,}*

Summary

Initial submission: Received : 9/4/2023

Scientific editor: Judith Nicholson

First round of review: Number of reviewers: 2
Revision invited : 10/14/2023
Revision received : 10/25/2023

Second round of review: Number of reviewers:
Accepted :

Data freely available: Yes

Code freely available: Yes

This transparent peer review record is not systematically proofread, type-set, or edited. Special characters, formatting, and equations may fail to render properly. Standard procedural text within the editor's letters has been deleted for the sake of brevity, but all official correspondence specific to the manuscript has been preserved.

Referees' reports, first round of review

Reviewer #1: In this manuscript, the authors tried to explore the genetic associations with protein-protein ratios that can uncover biologically relevant links between two or more proteins based on their shared genetic and non-genetic variance. The thought is interesting, and methodology they offered is potentially useful. Yet, there are problems or issues need to be address.

1. Data processing and Quality Control:

According to the description, the authors took quality control steps to handle sample data with a significant amount of missing protein values and proteins missing in many samples. This is a common quality control procedure to ensure the data analyzed is of high quality. Have you considered any other appropriate preprocessing and quality control steps? For instance, were outliers and potential outliers addressed?

2. Model Selection:

* When conducting GWAS or other statistical association analyses, were all relevant covariates considered? For example, potential confounding factors like age and gender.

* When using a linear model, were the model assumptions validated, such as linearity, independence, constant variance, and normality?

3. Statistical Interpretation of Ratios:

* Despite the article explaining that ratios are statistically considered for power, and this indeed has some theoretical support, the use of ratios may introduce new variability or noise. Particularly, when both protein levels are very low, ratios can become unstable. Has this potential instability been taken into account?

* For the association between ratios and genetic variation, are there additional analyses to explain or validate the significant p-gain observed in the ratios?

4. Biological Interpretation:

* Are there known biological explanations or evidence supporting the relationships between proteins mentioned in the article, especially when their associations were identified in blood? Are there further experiments or studies to validate these statistical associations?

* In some of the mentioned ratios in the article, the proteins involved may have multiple functions or be associated with multiple pathways. How was it determined that these ratios truly represent the mentioned biological processes rather than other potential pathways or functions?

5. Other Data Sources:

* The article references multiple external databases and tools such as STRING and IPA databases. What is the reliability and accuracy of these data sources? Were the limitations or biases of these databases considered when using them? For example, information in IPA and STRING databases may be based on specific experimental conditions or cell types, which might not align perfectly with the context of this study. How about the accuracy and completeness of other external resources or tools mentioned in the article, like CytokineLink? Are there known limitations or biases associated with them?

6. Robustness of Conclusions:

* Were sensitivity analyses or other analyses conducted to validate the robustness of the main conclusions?

* Have the major findings in the article been validated or replicated in other independent datasets?

Above all, the article would benefit from further refinement and elaboration in areas such as data preprocessing and quality control, model selection, statistical interpretation of ratios, biological interpretation, and the utilization of external data sources. Additionally, conducting sensitivity analyses to validate the robustness of the primary conclusions and replicating key findings in independent datasets is recommended to enhance the credibility and scientific significance of the article.

Reviewer #2: This paper presents interesting analysis regarding use of ratio of proteins to identify new

pQTL. It is an impressive single-author effort for carrying out a fairly comprehensive data analysis. The author has been able to show that use of ratios can identify new pQTLs, but the additional values of such pQTLs, which can have complex interpretation, for identification of drug targets in a more rigorous causal-inference type framework is lacking. Overall, I feel it is a good demonstration paper that ratios may have some interesting genetic signals, but the downstream implications are not clear. I only have a few major comments for this paper.

1) The authors present some statistical models under which the use of a ratio may give improved power for the detection of pQTLs. For correlated traits (proteins in this example), one can potentially use class of multivariate association tests for improving the power for detection of genetic signals. If that is the main goal, it is not clear to me why only look at ratios, and not consider other types of multivariate/bivariate genetic association tests. I feel the two scenarios the author come up with under which the use of ratios can increase power of association tests are not necessarily the most interesting or likely cases. For example, the authors argue ratios could pick up pQTL which affects some hidden factors shared across proteins, like platelet counts. But I would think the goal of pQTL analysis should be to really detect more protein specific effects so that those could be interpreted better subsequently, e.g in MR analysis.

2) Also, regarding shared effects, I wonder if the author has considered adjusting proteomic data using genomic surrogates, e.g those could be captured by using SVA analysis of proteomic data, which could capture some hidden underlying biological processes (e.g cell type counts). In the analysis of SomaScan data in the ARIC study, Zhang et al (Nature Genetics, 2022) showed that adjusting/controlling for hidden factors can lead to substantial improvement in pQTL detection across multiple populations.

3) While I do find the author has convincingly shown that more pQTLs can be discovered through using ratios of proteins, I feel downstream applications of these complex rPQTLs effects are not as clear or convincing. There is clearly no MR-type causal inference framework as there is for the interpretation of association for simple cis-pQTLs with complex traits. The author did build some stories around some findings, but they are speculative in nature and I think the paper would benefit if it used more formal framework. such as MR, to really point out which proteins are likely to be causal and thus potential drugs targets, vs which are just tagging along in a correlated network.

4) The entire emphasis of the paper, including literature review is the Olink platform. But a lot of pQTL data are also generated using platforms like SomaScan. I would be curious to see some evidence whether the rPQTLs are platform specific and if it is what it means. Of course these could be challenging due to very different contents of the platforms, but investigating this issue even in a limited basis can lead to better interpretation of the findings.

Authors' response to the first round of review

Reviewer #1

In this manuscript, the authors tried to explore the genetic associations with protein-protein ratios that can uncover biologically relevant links between two or more proteins based on their shared genetic and non-genetic variance. The thought is interesting, and methodology they offered is potentially useful. Yet, there are problems or issues need to be address.

Response: We thank the reviewers for their tome, interest, and suggestions, and hope to have answered all open points satisfactorily in the following.

1. Data processing and Quality Control:
- 2.

According to the description, the authors took quality control steps to handle sample data with a significant amount of missing protein values and proteins missing in many samples. This is a common quality control procedure to ensure the data analyzed is of high quality. Have you considered any other appropriate preprocessing and quality control steps? For instance, were outliers and potential outliers addressed?

Response: The data processing and quality control was conducted by the UKB PPP consortium. The Olink

data provided by UK Biobank is identical to what was used in the consortium's pQTL study (Sun et al.). By closely following data processing and quality control steps taken by the UKB PPP consortium we assure the comparability of our results with their study. For that reason, we have not considered any other preprocessing and quality control steps.

To be clear, the QC steps performed by the UKB PPP consortium include outlier sample detection and removal, removing data with QC warnings or assay warnings, removing likely sample swaps, batch corrections, etc.

Under the tab "Resources" on the UKB page the reader can access all relevant QC and data normalization documents for the Olink data (Figure X1 below). Similar documents are available for the genomics data. We reference these documents in the "Methods section" under "Data sources".

Contrary to the reviewer's understanding, there was not a significant amount of missing protein values and proteins missing in many samples. According to the Olink analysis report, the number of datapoints that passed QC was 97.6% and above, depending on the panel (Table X1 below).

To make this point clear to the reader, we added the following paragraph to the main manuscript: "Quality control of the Olink data was performed by the UKB PPP prior to sharing of the data with UK Biobank. These QC steps include outlier removal and removal of samples of low quality. The number of datapoints that passed QC was above 97.6% for all Olink panels (see methods)."

biobank^{uk} Index Browse Search Catalogues Downloads Login Help

Data-Field 30900

Description: Number of proteins measured
 Category: Biological samples > Blood assays > Proteomics > Protein biomarkers

Participants	52,748	Value Type	Integer, number of proteins	Sexed	Both sexes	Debut	Jan 2023
Item count	55,001	Item Type	Records	Instances	Defined (4)	Version	Feb 2023
Stability	Accruing	Strata	Derived	Array	No	Cost Tier	d2 o2 s2

Data | **3 Instances** | **Notes** | **1 Record Table** | **0 Related Data-Fields** | **12 Resources**

Preview Name	Res ID
Olink Analysis Report	4655
Olink Explore 1536 - FAQ	4657
Olink data normalisation strategy	4656
Olink proteomics data	4654
Quality control of olink NPX dataset	4658
olink assay	1013
olink assay version	1014
olink assay warning	1015
olink batch number	1016
olink limit of detection	1017
olink panel lot number	1018
olink processing start date	1019

Enabling scientific discoveries that improve human health

Figure X1: UKB Olink resource page <https://biobank.ndph.ox.ac.uk/showcase/field.cgi?id=30900>. This page is freely available to the user and contains all relevant QC documents and technical reports.

2.1 QC summary

Olink Panel	Samples passed QC	Samples passed QC (%)	Datapoints passed QC	Datapoints passed QC (%)
Cardiometabolic	54523 / 58369	93.41	21152616 / 21538161	98.21
Inflammation	54281 / 58363	93.01	21011339 / 21477584	97.83
Oncology	54153 / 58366	92.78	20961212 / 21478688	97.59
Neurology	54133 / 58366	92.75	20913874 / 21420322	97.64

Table X1: QC summary, table extracted from the Olink data QC report.

2. Model Selection:

* When conducting GWAS or other statistical association analyses, were all relevant covariates considered? For example, potential confounding factors like age and gender.

Response: We followed the statistical analysis methods employed by Sun et al. Note that Sun et al. followed in turn general practice in the proteomics and metabolomics GWAS field, i.e. using age, sex, and the first ten genoPCs as covariates. We therefore believe to have considered all relevant covariates. We specify these details in the "Methods section" under "GWAS analysis".

* When using a linear model, were the model assumptions validated, such as linearity, independence, constant variance, and normality?

Response: We follow common practices in the proteomics and metabolomics GWAS field. In particular, by using inverse-normal scaled concentrations as dependent variables we assure normality of the dependent variables. In general, model assumptions are as valid as they are for any of the other GWAS with UKB data, which we believe have been proven sufficient for the purpose of pQTL studies in the past.

3. Statistical Interpretation of Ratios:

* Despite the article explaining that ratios are statistically considered for power, and this indeed has some theoretical support, the use of ratios may introduce new variability or noise. Particularly, when both protein levels are very low, ratios can become unstable. Has this potential instability been taken into account?

Response: We agree that divisions by near zero values can result in very large values (outliers) and consequently over-estimated significance levels. However, as we inverse-normal scale after taking the ratios, this kind of instability should be controlled for. Furthermore, we require conservative Bonferroni significance and focus on replicated associations, which controls the level of spurious signals generated by noise. We therefore do not think that this is an issue.

* For the association between ratios and genetic variation, are there additional analyses to explain or validate the significant p-gain observed in the ratios?

Response: In a previous paper [ref 11: Petersen et al. On the hypothesis-free testing of metabolite ratios in genome-wide and metabolome-wide association studies. BMC Bioinformatics 13, 120 (2012)] we performed a theoretical analysis on ratios and showed that a p-gain of 10 is the equivalent of a nominal p-value for a single test, in other words, a p-gain of 10 is expected to be observed by chance in 5% of the cases when ratios between two random proteins are tested.

We require conservative Bonferroni significance for the p-gain by multiplying the nominal p-gain significance level ($p\text{-gain} > 10$) by the number of tests. In addition, we require replication in an independent cohort (the non-Caucasian samples in UKB). We believe that these are stringent criteria that assure the validity of the reported p-gains.

4. Biological Interpretation:

* Are there known biological explanations or evidence supporting the relationships between proteins mentioned in the article, especially when their associations were identified in blood? Are there further

experiments or studies to validate these statistical associations?

Response: There are many known biological explanations or evidence supporting the relationships between proteins. Basically, every protein-protein relationship identified by String (i.e. those with a score of 999 in Table ST4) provide such evidence and support a relationship between two proteins in a ratio. Further examples are the cytokine – cytokine receptor pairs and specific examples that we discuss in the paper, such as the association of the ratio of PCSK9 and LDLR at the APOE locus, where it is established that the former inhibits the latter.

Associations identified in blood may indeed reflect interactions that are happening in another organs, with the respective proteins being leaked or excreted into the blood stream. This is a general caveat to all blood-based proteomics analyses. In the introduction we refer the reader to our review paper (Figure X2) where we discuss these and other aspects of blood based pQTL studies in detail [ref 1: Suhre, McCarthy, Schwenk; Genetics meets proteomics: perspectives for large population-based studies, Nat. Rev. Genet. 2020].

* In some of the mentioned ratios in the article, the proteins involved may have multiple functions or be associated with multiple pathways. How was it determined that these ratios truly represent the mentioned biological processes rather than other potential pathways or functions?

Response: As for all GWAS, our study is hypothesis generating by nature. From the association data alone, it cannot be determined that these ratios represent the mentioned biological processes rather than other potential pathways or functions. The fact that proteins may have multiple functions or be associated with multiple pathways is a caveat that needs to be considered in the interpretation of the rQTLs on a one-by-one basis.

To make this point clearer, we added the following paragraph to the caveat section in the discussion: “As for all GWAS, our study is hypothesis generating by nature. An rQTL suggests a potential relationship between the two proteins in the ratios, plus potentially a third protein encoded at the genetic locus in the case of trans-rQTLs. These relationships can be of different nature, ranging from a direct physical interaction over indirect regulatory interactions to broad shared non-genetic factors. Also, proteins may have multiple functions or be associated with multiple pathways, which needs to be considered in the interpretation of the rQTLs on a one-by-one basis”.

5. Other Data Sources:

* The article references multiple external databases and tools such as STRING and IPA databases. What is the reliability and accuracy of these data sources? Were the limitations or biases of these databases considered when using them? For example, information in IPA and STRING databases may be based on specific experimental conditions or cell types, which might not align perfectly with the context of this study. How about the accuracy and completeness of other external resources or tools mentioned in the article, like CytokineLink? Are there known limitations or biases associated with them?

Response: String and IPA are current state-of-the-art databases for protein interactions. They incorporate data from many other and more specific databases. CytokineLink is a compilation of published cytokine – cytokine receptor interactions and as such represents the current knowledge of the field. These databases have their limitations and shortcomings, but they represent the best available resources to date. We are not aware of any shortcoming that limit the use of these database specifically for our purposes. Furthermore, we used random sampling to avoid bias in the calculation of the enrichment and report the standard deviation of the sampling process. We therefore do not feel that there are specific limitations or biases that we should report.

6. Robustness of Conclusions:

* Were sensitivity analyses or other analyses conducted to validate the robustness of the main conclusions?

Response: We believe that our use of a conservative Bonferroni significance levels in conjunction with an independent replication cohort assures the robustness of our main conclusions.

* Have the major findings in the article been validated or replicated in other independent datasets?

Response: Yes – we used the samples from the non-Caucasian UKB participants for replication. Above all, the article would benefit from further refinement and elaboration in areas such as data

preprocessing and quality control, model selection, statistical interpretation of ratios, biological interpretation, and the utilization of external data sources. Additionally, conducting sensitivity analyses to validate the robustness of the primary conclusions and replicating key findings in independent datasets is recommended to enhance the credibility and scientific significance of the article.

Response: This is a summary of the earlier points made by the reviewer. We believe that our previous responses answer all of these issues.

Stopped here

Reviewer #2:

This paper presents interesting analysis regarding use of ratio of proteins to identify new pQTL. It is an impressive single-author effort for carrying out a fairly comprehensive data analysis. The author has been able to show that use of ratios can identify new pQTLs, but the additional values of such pQTLs, which can have complex interpretation, for identification of drug targets in a more rigorous causal inference type framework is lacking. Overall, I feel it is a good demonstration paper that ratios may have some interesting genetic signals, but the downstream implications are not clear. I only have a few major comments for this paper.

Response: We like thank the reviewers for their time, interest, and suggestions, and hope to have addressed all concerns in the revision.

1) The authors present some statistical models under which the use of a ratio may give improved power for the detection of pQTLs. For correlated traits (proteins in this example), one can potentially use class of multivariate association tests for improving the power for detection of genetic signals. If that is the main goal, it is not clear to me why only look at ratios, and not consider other types of multivariate/bivariate genetic association tests. I feel the two scenarios the author come up with under which the use of ratios can increase power of association tests are not necessarily the most interesting or likely cases. For example, the authors argue ratios could pick up pQTL which affects some hidden factors shared across proteins, like platelet counts. But I would think the goal of pQTL analysis should be to really detect more protein specific effects so that those could be interpreted better subsequently, e.g in MR analysis.

Response: We agree with the reviewer that a more general class of multivariate association tests can further improve the power for detection of genetic signals.

However, we feel that this is a direction of research that goes beyond the scope of our current study. We chose to stick to simple ratios here, as this is a concept that translates into traits that most metabolomics and proteomics researchers are already familiar with, like ratios of urine metabolites with creatinine to account for sample dilution (an example of shared non-genetic variance) and ratios between substrate and product pairs as a proxy for a reaction throughput (an example of shared genetic variance).

We agree that the concept of ratios can (and should) be extended to multivariate associations in the future, possibly including approaches like variable selection and structural modelling, but doing so requires extensive further research and development.

To address this point, we added the following statement to the conclusion:

“Further theoretical development and generalization of the concept of ratios to more general classes of multivariate association tests may further improve the power for detection of genetic signals.”

2) Also, regarding shared effects, I wonder if the author has considered adjusting proteomic data using genomic surrogates, e.g those could be captured by using SVA analysis of proteomic data, which could capture some hidden underlying biological processes (e.g cell type counts). In the analysis of SomaScan data in the ARIC study, Zhang et al (Nature Genetics, 2022) showed that adjusting/controlling for hidden factors can lead to substantial improvement in pQTL detection across multiple populations.

Response: We agree with the reviewer that adjusting for hidden structures in the genomic and/or proteomic data is a way of increasing the power of pQTL studies that conceptually intersects with our approach of using ratios. Indeed, similar to Zhang et al., we observed an increase in statistical power in

our previous pGWAS with Somalogic by adjusting for the first principal component (PC1) of the proteomics data. PC1 could be interpreted as a signature of “white blood cell lysis” and be proxied by HSP90 protein levels, which is effectively equivalent to using ratios with HSP90.

We added the following paragraph to section “Why do ratios work and what do they represent?” to acknowledge this approach:

“It is interesting to note in this context that adjusting for hidden factors by using genomic or proteomic surrogates can also lead to substantial improvement in pQTL detection. [ref Zhang et al.]”

3) While I do find the author has convincingly shown that more pQTLs can be discovered through using ratios of proteins, I feel downstream applications of these complex rPQTLs effects are not as clear or convincing.

Response: We provide and discuss an extensive set of downstream applications in the paper, including:

1. The identification of proteins that associate in a ratio with an established drug target and that could therefore constitute for example interactors with the target, biomarkers for target engagement, or identify the presence of shared variance that may be confounding experiments related to drug efficacy (Table 1);

2. The discovery of 39 previously unidentified cis-pQTLs that validate the target specificity of the affinity binder and thereby render conclusions regarding future associations of these Olink protein readouts more reliable (Table 2);

3. An example of how rQTLs can support the identification of a causal gene at a clinically relevant locus (Crohn’s disease, Table 3). Similar arguments can be made for many of the reported rQTLs, which we provide as a resource;

4. We report an additional 25% of genetic loci by using a purely computational approach (Table ST6, filter column CB on “FALSE”). The UKB PPP consortium invested over \$20 million to generate the Olink data with the central aim of generating as many pQTLs as possible for hypothesis generation, implying a concrete monetary value for these loci and providing a strong motivation for the wider use of the approach.

5. The identification of 322 new loci that overlap with a clinical GWAS catalogue hit and that were not identified in the UKB PPP study (Table ST7). These associations can support the development of new drug targets, as we discuss in the example of the STK1 / USP8 ratio and its association with Alzheimer’s disease.

Taken together, we believe that these points represent a strong and comprehensive list of possible downstream applications of rQTLs.

There is clearly no MR-type causal inference framework as there is for the interpretation of association for simple cis-pQTLs with complex traits. The author did build some stories around some findings, but they are speculative in nature and I think the paper would benefit if it used more formal framework, such as MR, to really point out which proteins are likely to be causal and thus potential drug targets, vs which are just tagging along in a correlated network.

Response: We agree that “There is clearly no MR-type causal inference framework as there is for the interpretation of association for simple cis-pQTLs with complex traits.” We therefore do not understand why the reviewer suggests that “the paper would benefit if it used more formal framework, such as MR”. We believe that causal inference with ratios is a field of great interest for future research on MR techniques, but it is beyond the scope of our current study to develop such a framework. We hope that our paper will inspire such research, which should be accessible to others as we freely share all summary statistics.

4) The entire emphasis of the paper, including literature review is the Olink platform. But a lot of pQTL data are also generated using platforms like SomaScan. I would be curious to see some evidence whether the rPQTLs are platform specific and if it is what it means. Of course these could be challenging due to very different contents of the platforms, but investigating this issue even in a limited basis can lead to better interpretation of the findings.

Response: We respectfully disagree with the assessment that the entire emphasis is on the Olink platform. References 2,4, and 5 are studies reporting GWAS with the SomaScan platform. We actually conducted the first GWAS at scale using the SomaScan platform (ref5: Suhre et al. Nat. Comms 2017) where we already reported selected associations with ratios. Our experience from that study prompted the present investigation. We mention this in the introduction:

“Previous GWAS with proteomics suggest that Gaussian graphical models (GGMs) built from partial correlations and ratios between protein levels can reveal biologically relevant protein-protein interactions (ref 5), but the approach has never been tested at scale.”

We also previously showed that significant p-gains with ratios are not platform-specific and also work for metabolites and other quantitative traits. We specify this in the introduction:

“We and others previously developed analysis strategies for GWAS with metabolomics data (ref 7,8), a field that is similar in many ways to that of pQTL studies. In particular, we showed that partial correlations between metabolites can reconstruct metabolic networks (ref 9,10) and that the hypothesisfree testing of all ratios between metabolites can substantially strengthen the association signals, in several cases elevating genetic loci out of the background noise (ref 11,12).”

Referees' reports, second round of review

Reviewer #1: Comments enter in this field will be shared with the author; your identity will remain anonymous.

As all the results are from Bioinformatics Analysis, I would like to see some experimental evidence for the regulational networks the author discovered (as shown in Fig 6 or Fig 7).

Reviewer #2: I m mostly satisfied with the responses. However, the author should acknowledge that the lack of availability of a causal inference framework somewhat limits the interpretation of the findings. The author in the response talks about partial correlations to characterize casual networks. However, partial correlation between observed proteins/metabolites does not lead to casual networks. First, partial correlation can lead to spurious relationships between variables due to "collider" effects. Second, a causal network analysis will require some kind of "randomized" evidence, such as those from pQTLs. It is a complex question and I don't expect the MR framework to be developed in the paper. But the limitation of interpretation of results in the absence of a formal framework needs to be pointed out.

Authors' response to the second round of review

Reviewer #1:

As all the results are from Bioinformatics Analysis, I would like to see some experimental evidence for the regulational networks the author discovered (as shown in Fig 6 or Fig 7).

Editor's decision letter: We would not expect experimental validation as suggested by reviewer 1, but would recommend adding more support for the analyses in fig 6 and 7 from the literature and textually.

Response: We followed the editors suggestion and added the following literature evidence to the respective sections:

“For instance, it has been shown that silencing of NFATC1 results in phosphorylation of FOXO1 and thereby plays a role in cell differentiation 33. Expression of NFATC1 and nine of the proteins in its ratio (AXIN1, BCR, CASP2, CD69, EIF4G1, FADD, IRAK1, LBR, PTPN6) is enriched in leukemia cells (FDR=0.003) 34. IRAK1 is an emerging therapeutic target in hematologic malignancies, and it has been suggested that

IRAK kinases participate in regulatory interactions with FADD 35. Indeed, FADD has been shown to physically interact with IRAK1 36, lending further experimental support to this rQTL-derived network and its role in T-cell development.”

and

“An interaction between these two cytokines has experimentally established by bidirectional immunoligand blotting 40”

Reviewer #2:

I m mostly satisfied with the responses. However, the author should acknowledge that the lack of availability of a causal inference framework somewhat limits the interpretation of the findings. The author in the response talks about partial correlations to characterize casual networks. However, partial correlation between observed proteins/metabolites does not lead to casual networks. First, partial correlation can lead to spurious relationships between variables due to "collider" effects. Second, a causal network analysis will require some kind of "randomized" evidence, such as those from pQTLs. It is a complex question and I don't expect the MR framework to be developed in the paper. But the limitation of interpretation of results in the absence of a formal framework needs to be pointed out. Editor's decision letter: We would [...] recommend addressing the discussion points raised by reviewer 2 in the text.

Response: We agree with the reviewer and added the following caveat to the discussion:

“We also acknowledge that there is presently no established causal inference framework which somewhat limits the interpretation of the findings.”